# Effect of the Sway Bed on Autonomic Response, Emotional Responses, and Muscle Hardness in Children with Severe Motor and Intellectual Disabilities: A Pilot Study

**DOI:** 10.3390/healthcare10112337

**Published:** 2022-11-21

**Authors:** Mitsuki Ozaki, Jun Murata, Katsuya Sato, Goro Tanaka, Akira Imamura, Ryoichiro Iwanaga

**Affiliations:** 1Department of Occupational Therapy Sciences, Graduate School of Biomedical Sciences, Nagasaki University, 1-7-1 Sakamoto, Nagasaki 852-8520, Japan; 2Isahaya Special Education High School, 1670-1 Masaki, Isahaya 854-0084, Japan

**Keywords:** severe motor and intellectual disabilities, autonomic response, emotional response

## Abstract

This cross-sectional study aimed to examine the effects of being swayed in a sway bed on children with severe motor and intellectual difficulties by examining potential differences in their autonomic and emotional responses, as well as their muscle hardness, and by comparing them with “a control condition without any stimulation”. Children’s heart rate variability, rectus femoris hardness, and passive hip abduction range of motion (ROM) were measured in two experimental conditions, differentiated by the presence of a 5-min sway stimulus. In each condition, the children’s faces were video-recorded and retrospectively rated subjectively by their homeroom teacher concerning the visible expression of eight emotions. Significant intervention-related effects were observed on the heart rate variability and the “Relax” item of the emotional response indicators but not on muscle hardness or hip ROM. Our findings provide evidence that using a motorized sway bed can promote relaxation in children with severe motor and intellectual disabilities by influencing their autonomic response.

## 1. Introduction

“Severe motor and intellectual disabilities (SMID)” in rehabilitation medicine characterizes chair- or bedridden individuals with an intelligence quotient (IQ) of ≤35 [1]. Individuals with SMID struggle with various health problems, such as dysphagia, seizures, secondary musculoskeletal disorders, and chronic breathing problems, along with communication and cognitive difficulties, are associated with emotional and behavioral impairments [2,3]. In children with SMID, motor ability has been linked to the quality of life (QOL) [4], with dysfunction inversely associated with QOL in affected children and their families [5,6]. Dystonia, another common feature of SMID, is known to affect motor function. Therefore, therapeutic exercises need to focus on the control of muscle tone [7].

Disability rehabilitation deploys relaxation techniques, such as postural positioning and muscle stretching, to help children with SMID learn how to control their muscle tone [8,9]. Such techniques are meant to induce relaxation under physical and emotional stress. When followed, the sympathetic activation of the “fight-or-flight” response is suppressed, allowing individuals to react in a calmer and more composed manner [10]. The relaxation methods decrease sympathetic function and increase parasympathetic function, allowing remodeling to a favorable autonomic balance [11]. While their effectiveness varies, relaxation techniques have been reported to efficiently relieve physical and mental stress while reducing pain and anxiety levels [12].

For countless centuries, rocking has been used to promote sleep and relaxation in infants [13]. Rocking also influences sleep among adults by promoting relaxation [14]. It is considered to reduce psychological tension, as similar effects have been observed after performing similar activities involving swaying or oscillatory motion. For example, in children with cerebral palsy, horseback riding has been found to reduce muscle tone [15] and to correct imbalances in autonomic activity [16], while whole-body vibration reportedly improves muscle tone, joint range of motion (ROM), and ambulatory function [17]. For children with SMID, an intervention whereby the children were rocked and sung to while being hugged was confirmed to promote relaxation effectively [18].

The sway bed is a welfare device that was created at the request of parents to provide muscle tension relief and psychological relaxation to children with SMID. Although it was created for this purpose, its effectiveness has not yet been fully verified. In a previous study, we reported that sway in children with SMID using such a device improved their joint motility, as well as increased their pleasure and satisfaction, as rated by the observers [19]. Our findings suggested that this population may benefit from such sway stimulation by alleviating emotional response and muscle hardness.

However, although previous studies have investigated changes when sway stimulus was given, the studies were limited, because the data were compared before and after the intervention, and no comparison was made with cases where sway stimulus was not given (control condition). Therefore, the purpose of this pilot study was to examine the relaxation effect of sway stimulus using a sway bed for autonomic and emotional responses, as well as muscle hardness in children with SMID.

## 2. Materials and Methods

### 2.1. Participants

Our participants were 10 students attending a special support school who were diagnosed with SMID (median age, 15 years; Table 1). Bedridden children who were unable to sit up or move by themselves were included in the study. Children with the following conditions were excluded from the study: Gross Motor Function Classification System (GMFCS) levels I–IV, aspiration pneumonia, respiratory disease, heart disease, susceptibility to motion sickness, and sensitive skin (to avoid irritation from electrocardiogram [ECG] electrodes). Initially, 13 children were included in the study; however, three children with GMFCS level IV were excluded, resulting in a final total of 10 participants. We obtained informed consent after providing children, parents, and their homeroom teacher with an overview of the experiment, explaining that their participation was voluntary and could be discontinued at any time if the children appeared distressed or physically unwell.

### 2.2. Location

The experiment was conducted in a private room used by students at the special support school. This room was kept quiet, and an ambient temperature of 26–28 °C was maintained to ensure consistent environmental conditions for all the children.

### 2.3. Sway Bed

Figure 1 shows the electric sway bed (RHYTHM BED Standard YB, IQuark, Fukuoka, Japan). The dimensions of the electric sway bed were 1900 × 900 × 500 mm (length × width × height), powered by an electric motor. It is a motorized device and was set to sway the child vertically by ±2 cm for a total of 4 cm. The device has seven sway speed settings. Based on the findings from a previous study [19] and the children’s tolerance at the time of the test run, we decided to use the level 2 (0.64 Hz) sway speed in our study.

### 2.4. Study Outcomes

#### 2.4.1. Autonomic Response

A pair of electrodes (BlueSensor VL, VL-00-S, Ambu, Ballerup, Denmark) was attached to the children’s chests for the ECG measurements. ECG signals were recorded continuously during each 5-min block using a heart rate variability (HRV) analyzer (Check My Heart V3.0: DailyCare BioMedical, Chungli, Taiwan). A frequency domain analysis was performed using the manufacturer’s software. ECG signals were decomposed using a fast Fourier transform to isolate activity in the low- (LF) and high-frequency (HF) bands (LF, 0.04–0.15 Hz; HF, 0.15–0.40 Hz). The LF/HF ratio was also calculated from the obtained data.

#### 2.4.2. Emotional Responses

Each homeroom teacher was asked to evaluate the children’s facial expressions (showing pleasure or displeasure) during the experiment using an observation questionnaire from a previous study [19]. The homeroom teacher was asked to rate eight items: “wakefulness (arousal), enjoyment, physical activity, relaxation, satisfaction, anger, depression or sadness, and anxiety or fear” using a 10-point scale. In rehabilitation, such questionnaires are often used to evaluate dyspnea and fatigue subjectively and numerically [20,21].

Two digital cameras were fixed to the railing of the rocking bed to record the children’s facial expressions during the experiment. The children’s facial expressions in the control condition (stimulus (none)) and in the sway stimulus condition (stimulus (sway)) were recorded by video. For the evaluation of the facial expressions, the homeroom teacher was presented with a randomized time series of videos from the control and sway stimulus conditions after the experiment before completing an observation questionnaire.

#### 2.4.3. Muscle Hardness

This outcome was measured in the left and right rectus femoris using the NEUTONE Muscle Hardness Tester (TDM-NA1: TRY–ALL Corporation, Chiba, Japan). Muscle hardness testers are easily portable and used to assess muscle relaxation [22]. Readings were obtained while the children were lying in a supine position on the sway bed, with their hips at 45° and knees at 90° flexion, by pressing the device down (perpendicular to the skin) at a designated position on their thigh. This site was determined individually for each child by measuring their femur length using a tape measure, followed by marking their thigh at one-third of the length from the groin using a water-soluble pen. Data were recorded as the average of three readings. Furthermore, the distance between the two knees during the passive opening and closing of the hip joint was measured, while the student’s heel position was fixed to the bed with the hip joint flexed at 45° and the knee joint flexed at 90° with the student in a supine position on the bed (Figure 2).

### 2.5. Experimental Protocol

The study employed a cross-sectional experimental design. Experiments were conducted at least 2 h after the children ate their last meal to prevent confounding effects on autonomic activity. To start the experiment, the children laid in a supine position on the sway bed; their limbs were not restrained, and they were free to rest their arms and legs in a position they found comfortable. Each experiment was attended by the child’s homeroom teacher to help prevent excess anxiety and make necessary decisions if they appeared unwell. Prior to experiment initiation, the researcher checked with the teacher that the child had not recently been in any stressful situations. To keep psychological apprehension to a minimum, the sway bed was brought to the school 3 months before for test runs and kept in a separate room where every student was free to use it.

The preliminary experiment was conducted once for all children in the presence of their homeroom teacher. We checked to see if the children disliked the swaying of the sway bed and whether or not they had any physical problems.

The experiment consisted of two protocols (Figure 3). The control conditions were Rest 1 (5 min), Calm (5 min), Stimulus (none) (5 min), and Rest 2 (5 min) (total of 20 min). The sway stimulus conditions were Rest 1 (5 min), Calm (5 min), Stimulus (sway) (5 min), and Rest 2 (5 min) (total of 20 min). Both Rest and Calm were states where the children laid in a supine position on the sway bed, but Rest was the time before and after the experiment when the children could relax their feelings and body, while Calm was the time when no sway stimulus was applied. We provided 5 min for each of these states to minimize the physical burden of the experiment on the children, taking into consideration their state during the preliminary experiments. The fluctuations in heart rate were measured during the Calm segment and Sway segment of each protocol. In addition, we measured the muscle hardness and hip ROM before and after the second Calm segment under control conditions and before and after the Sway segment under sway stimulus conditions.

The order in which the two protocols were administered to the children was randomly assigned. The criteria for discontinuing the experiment were as follows: whether the child or guardian indicated their wish to stop the experiments and when the child exhibited symptoms, such as facial expressions showing displeasure, pale face, vomiting, or abnormal nystagmus.

### 2.6. Data Analysis

The measurement data obtained were classified for each condition using the data from the Calm segment as the baseline value. Then, (1) the values of the Calm and Stimulus segments for each condition, (2) the amount of change between conditions for each measured item, and (3) the mean value for each condition of the observation questionnaire administered to the homeroom teachers were analyzed using Wilcoxon’s signed-rank test.

In addition, the effect size (r) was calculated for each. The strength of the effect size was assessed as small (<0.3), moderate (0.3–0.49), or large (>0.5). SPSS Statistics for Windows, version 22, was used for the analyses (IBM Corp., Armonk, NY, USA). The level of significance was set at *p* < 0.05.

### 2.7. Ethical Considerations

This study was reviewed and approved by the ethics review board of Nagasaki University Graduate School of Biomedical Sciences (approval no. 20073103-2). Children, parents, and school officials provided informed consent after receiving relevant explanations and documentation from a study researcher. Steps were taken to ensure the children’s willingness to participate, and their ability to decide independently was considered; consent was obtained as they relayed their intentions to their parent or teacher using an eye-gaze input or switch-controlled device; they were informed that they could withdraw that agreement at any time.

## 3. Results

Table 2 shows the Medians (interquartile range) and the amount of change between conditions for each measured item before and after swaying under each condition. Wilcoxon’s rank-signed test revealed no significant differences in muscle hardness of the right and left rectus femoris muscles and hip displacement distance between pre and post of both conditions. There was no significant difference in LF/HF and HF in the control condition, but there was a significant difference in the sway stimulus condition between pre- and postintervention (*p* < 0.05). When the effect sizes pre- and postintervention were calculated, the results indicated that r = −0.89 for LF/HF and r = −0.89 for HF, both of which were large. Seven of the ten children (Children A, B, C, D, E, G, and J) showed a decrease in the LF/HF component and an increase in the HF component pre and post the sway stimulus condition compared to the control condition. In contrast, the LF/HF component increased, and the HF component decreased in the other three children (F, H, and I) pre and post the sway stimulus condition compared to the control condition. No consistent characteristics were observed in each group of seven children with confirmed responses and in three children without confirmed responses. Since the number of cases was too small to statistically test the differences in the children’s characteristics between the two groups, the factors influencing the differences in the autonomic responses by the sway stimulus condition between the two groups could not be clarified. Regarding the amount of change in the pre- and postintervention values of each measurement item between the conditions, no significant differences were observed in the muscle hardness of the right and left rectus femoris and in the hip opening distance. However, significant differences were observed in the amount of change in the LF/HF and HF components from pre- to postintervention between the conditions (both *p* < 0.05). When these effect sizes were calculated, the results showed that r = −0.85 for LF/HF and r = −0.73 for HF, both indicating a large effect size.

Table 3 shows the medians (quartiles) and statistics for each condition of the observation questionnaire administered to the homeroom teachers. Only the item “relaxation” was significantly different (*p* < 0.05), while the other items were not significantly different. Similarly, the effect size before and after the intervention was large for the “Relaxation” item, with r = −0.64.

**Table 2 healthcare-10-02337-t002:** Median (interquartile range) and the amount of change between conditions for each measured item before and after swaying under each condition.

Item	Control Conditions			Sway Stimulus Conditions			Amount of Change		
							Control Conditions	Sway Stimulus Conditions	Comparison between the Two Conditions
	Pre	Post	*p*	Pre	Post	*p*	Pre	Post	*p*
	Median (IQR)			Median (IQR)			Median (IQR)		
LF/HF	1.09 (0.5–1.3)	1.04 (0.6–1.8)	0.38	0.96 (0.8–2.6)	0.79 (0.4–1.8)	<0.005	0.25 (−0.3–−0.7)	−0.34 (−0.3–−0.7)	<0.007
HF	48.03 (42.8–63.2)	48.97 (35.4–61.1)	0.44	51.04 (27.2–55.5)	55.78 (34.6–69.9)	<0.005	−5.7 (−15.7–−13.7)	7.50 (4.4–15.8)	<0.01
Muscle hardness [right] (N)	33.5 (28.7–39.0)	33.00 (29.7–37.2)	0.79	37.00 (31.5–39.5)	36.50 (28.5–39.5)	0.73	−5.72 (−15.7–13.7)	7.50 (4.4–15.8)	0.62
Muscle hardness [left] (N)	35.0 (32.0–39.2)	36.00 (31.0–37.5)	0.36	37.50 (34.7–40.2)	36.00 (35.7–39.2)	0.70	−1 (−2.5–1.2)	0.5 (−1–2.2)	0.21
Hip range of motion (cm)	51.00 (28.0–62.7)	48.50 (26.1–56.7)	0.87	47.50 (33.3–65.2)	45.50 (35.2–58.2)	0.53	1.00 (−6.5–4.0)	0.75 (−5.2–3.5)	0.87

LF, low-frequency; HF, high-frequency, Pre: before control or stimulus condition, Post: after control or stimulus condition

**Table 3 healthcare-10-02337-t003:** Median (interquartile range) and test values of subjective emotion ratings by trial condition.

Item	Control Conditions	Sway Stimulus Conditions	*p*
	Median (IQR)	Median (IQR)	
Awakening/arousal	10 (10–10)	10 (10–10)	0.32
Enjoyment	5.5 (5–7)	7 (5–7)	0.29
Body movement	5 (2–6)	5 (3–5)	0.46
Relaxation	7 (4–8)	8 (7–10)	<0.04
Satisfaction	6.5 (5–8)	6.5 (5–7.5)	0.31
Anger	1 (1–1)	1 (1–1)	0.32
Depression/sorrow	1 (1–1)	1 (1–1)	0.31
Anxiety/fear	1 (1–1.3)	1 (1–1)	0.18

## 4. Discussion

The purpose of this study was to examine the changes in autonomic nervous activity associated with the use of sway beds, as well as the effects on emotional response and muscle hardness in children with SMID. No significant change was observed in muscle hardness, but in the frequency analysis of the ECG, the LF/HF component decreased, and the HF component increased as a result of the sway stimulus. In addition, in the observation questionnaire completed by the homeroom teacher, a significant difference was noted only in the “Relax” item, and the effect size was “Large”. Therefore, we believe that the sway stimulus of the sway bed affects the psychological state of children with SMID, and this effect may have been reflected in the changes in autonomic nervous activity.

### 4.1. The Impact of Sway Stimuli on Relaxation in Children with SMID

The LF component of HRV reflects sympathetic and parasympathetic modulation, whereas the HF component mainly reflects parasympathetic modulation. The LF/HF ratio represents a measure of the sympathetic–parasympathetic balance [23,24].

According to previous studies, massage therapy increased the HF component and decreased the LF/HF component. These results show that massage therapy is effective for psychological relaxation [25,26]. Another report indicated that, when healthy participants listened to quiet music or inhaled the vapor of bergamot oil for a certain period of time, there were significant changes to the LF, HF, and the LF/HF ratio, as well as a shifting of the autonomic nerve balance in the direction of parasympathetic dominance, causing a relaxation effect [27]. In addition, another study reported that applying various sway stimuli to adults, such as translation and rotation for 5 min using a swaying bed, resulted in the sway stimulus in the vertical axis causing a greater relaxing effect [28]. Similarly, in our study, the sway stimulus caused the LF/HF component to decrease and the HF component to increase. Therefore, it is possible that the sway stimulus applied by the sway bed promotes a psychological relaxation effect in children with SMID.

In contrast, the emotional response measured using the questionnaire indicated a significant effect for the “Relax” item, and the effect size was deemed “Large”. Therefore, we reasoned that the sway stimulus of the sway bed affected the psychological state of children with SMID and that its relaxation effect may have been reflected in the changes in the autonomic nervous system activity of these children.

### 4.2. The Impact of Sway Stimuli on Muscle Hardness in Children with SMID

We did not observe any improvements in muscle hardness of the left and right rectus femoris muscles and hip ROM. This suggests that the sway stimulus may not cause changes to the muscle hardness or ROM. However, the methodology of our study may have affected this result. One reason for the absence of change in the aforementioned measurements after the application of the sway stimulus may have been the fact that the sway stimulus was not applied long enough. We limited the sway stimulus to 5 min to minimize the physical burden on the children, considering their state during our preliminary observations. However, previous studies have reported that the application of a sway stimulus for 10 min led to an increase in joint ROM, a reduction in spasticity, and an improvement in gait function [17,19]. To examine the relationship between the sway stimulus, muscle hardness, and hip ROM in detail, one should consider prolonging the application of the sway stimulus in the future.

### 4.3. Limitations

This study has limitations concerning the sample size and age range. Especially, because of the small sample size of 10 individuals, it was not possible to examine the relationships between sex, age, disability type, and the degree of data change before and after the sway stimulus. As various causes of SMID are latent, we should examine the relationships between participant characteristics and data change in a large cohort. In addition, as the number of participants in this study was small, we were unable to clarify the differences in the characteristics of children in whom the LF/HF component decreased or the HF component increased due to sway stimuli, as well as in those in whom it did not. Future analyses using a larger sample size are needed to examine this issue. Furthermore, the wide age range of the participants in this study may have influenced the results. Future studies should include individuals with a narrower age range.

### 4.4. Introduction of Sway Bed Use in the Clinical Setting for Children with SMID

Although previous works have suggested that swaying promotes relaxation and sleep in infants [13] and affects adult sleep while promoting relaxation effects [14], only a few studies have investigated the impact of sway stimuli applied by sway beds on children with SMID. However, as reported in previous studies [15,16,17,18], applying a sway stimulus to children with SMID through the introduction of sway beds in schools, in the context of rehabilitation, or as part of leisure activities may result in a daily relaxation effect on children with SMID. To actually apply such sway bed policies in clinical settings, more detailed studies are needed in the future to examine the effect of the sway stimulus in children.

## 5. Conclusions

In this study, we examined the impact of sway stimuli applied using sway beds on autonomic nervous and emotional responses, as well as muscle hardness in children with SMID. This is one of a few studies that have compared the control and sway stimuli conditions for children with SMID. Although we were unable to observe any impact of the intervention on muscle hardness, we confirmed that sway stimuli affected the autonomic nervous and emotional responses in children with SMID. Future studies are needed to examine in detail the measurement indicators and the time and frequency for applying such sway stimuli.

## Figures and Tables

**Figure 1 healthcare-10-02337-f001:**
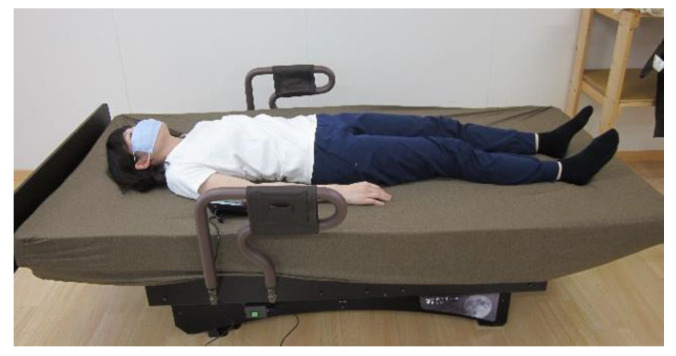
An electric sway bed (dimensions: 1900 × 900 × 500 mm (length × width × height)), powered by an electric motor, with seven sway speed settings. The motorized device was set to sway the student vertically by ±2 cm for a total of 4 cm. In this study, the level 2 (0.64 Hz) sway speed was used.

**Figure 2 healthcare-10-02337-f002:**
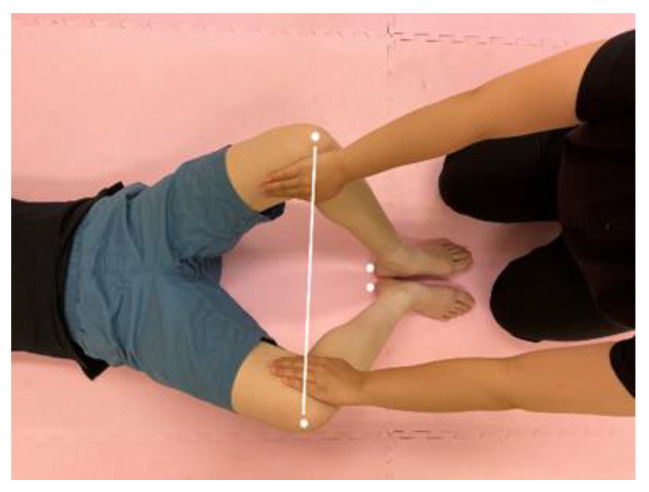
Measurement of hip range of motion. The children were asked to lie in a supine position, and the heel position was fixed to the bed with the hip joint flexed to 45° and the knee joint flexed to 90°. In that state, the hip range of motion was measured while performing the passive hip joint opening and closing exercise.

**Figure 3 healthcare-10-02337-f003:**
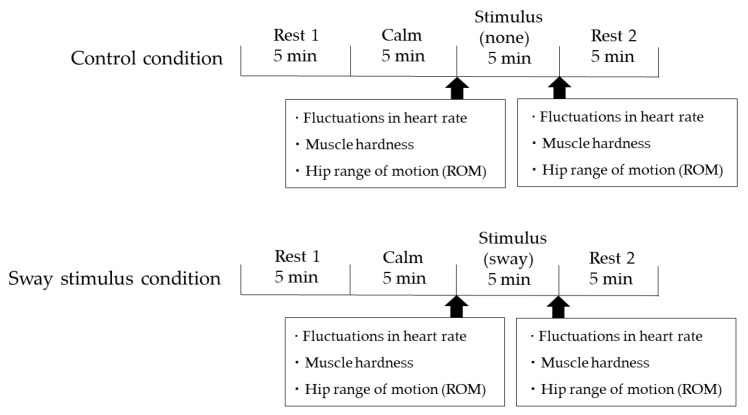
Details of each condition and timing of the measurements. The control conditions were Rest 1 (5 min), Calm (5 min), Stimulus (none) (5 min), and Rest 2 (5 min) (total of 20 min). The sway stimulus conditions were Rest 1 (5 min), Calm (5 min), Stimulus (sway) (5 min), and Rest 2 (5 min) (total of 20 min). The fluctuations in heart rate were measured during the Calm and Sway segments of each protocol. We measured the muscle hardness and the hip ROM before and after the second Calm segment under the control conditions, as well as before and after the Sway segment under the sway stimulus conditions.

**Table 1 healthcare-10-02337-t001:** Basic information concerning the participating children.

Case	Sex	Age	Bodyweight (kg)	Diagnosis	GMFCS ^1^ Level	Characteristics
A	Male	17 years and 9 months	38.0	Sequelae of an acute encephalopathy, Refractory epilepsy	V	No clear response when spoken to.
B	Male	16 years and 4 months	18.9	Cerebral palsy (athetosis type), Osteoarthritis, Periventricular leukomalacia, Kernicterus	V	No clear response when spoken to. Switches gaze to the direction of voice and smiles at times when spoken to.
C	Female	17 years and 1 month	37.2	Low birth weight, Cerebral palsy, Congenital cataract, Suspected Leigh syndrome	V	Shifts gaze, and moves left hand a little to make simple yes/no signals when spoken to. Smiles when comfortable and distorts face when uncomfortable.
D	Male	14 years and 2 months	18.7	Sequelae of an acute encephalopathy, Pneumonia, Epilepsy, Ulcerative colitis, Scoliosis, Dehydration	V	No clear response when spoken to. Switches gaze to the direction of voice and smiles at times when spoken to.
E	Male	13 years and 10 months	25.4	Very low birth weight, Cerebral palsy, Whooping cough, respiratory arrest, Bronchitis, Hip dislocation	V	No clear response when spoken to. Smiles when comfortable and sticks tongue out when uncomfortable.
F	Male	15 years and 5 months	32.4	Cerebral palsy, Lennox–Gastaut syndrome	V	No clear response when spoken to.
G	Male	17 years and 10 months	42.0	Ultralow birth weight, Epilepsy, Bronchopneumonia	V	Capable of simple expressions of intention, such as yes/no, by vocalizing “ahh.” Smiles when comfortable and distorts face when uncomfortable.
H	Male	18 years and 2 months	39.5	Cerebral palsy	V	Capable of simple expressions of intent, such as yes/no, through the slight movement of neck and nodding. Smiles when comfortable and distorts face when uncomfortable.
I	Male	8 years and 7 months	19.8	Lennox–Gastaut syndrome, Delayed motor development, Hip dislocation	V	No clear response when spoken to.
J	Male	8 years and 9 months	14.0	MCT8 deficiency, Hip dislocation	V	No clear response when spoken to. Smiles when comfortable and cries when uncomfortable.

^1^ GMFCS, gross motor neuron classification system.

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
