# Peer review of "Effect of the Sway Bed on Autonomic Response, Emotional Responses, and Muscle Hardness in Children with Severe Motor and Intellectual Disabilities: A Pilot Study"

_healthcare, 2022, doi:10.3390/healthcare10112337_

Round 1

Reviewer 1 Report

 The authors aimed in this study to examine the effects of being rocked in a motorized bed on children with severe motor and intellectual difficulties(SMID) by examining potential differences in their autonomic/emotional responses and  muscle tone. My concerns are as follows.

1.       In Abstract it is described “by comparing them with an “unrocked”  control group. However, the comparison seemed to have conducted between control and sway stimulus conditions. I also wonder whether there was any difference between “condition” in Table 2 and “group” in Table 3. It is necessary to describe the study design and results to be shown more clearly.

2.       The subjects of this study were 10 students attending a special support school who had been diagnosed with SMID, aged between 8 years and 7 months and 18 years and 2 months. I consider that this age range is too wide to examine the effect of the sway bed. The median age is better to be described.

3.       The authors showed the characteristics of 10 patients in Table 1. I consider that it would be easier for readers to understand their background characteristics that their summarized data are shown in a table.

4.       It is described in 2.4.2. Emotional responses “After the experiment, the homeroom teacher was given a random time series of the videos of the control and sway stimuli conditions to assess the children’s emotions based on the 8 items of the observation questionnaire and to evaluate them with a 10-point scale”. I wonder whether the assessment using a 10-point scale was reproducible and objective without any clear scale explanation. More detailed descriptions of the assessment of emotional response are necessary.

5.       The core findings of this study were the LF/HF component decreased, and the HF component increased as a result of the sway stimulus, in the frequency analysis of the electrocardiogram, and the subjective emotion rating of Relaxation was significantly larger under sway condition. Due to the wide variety and diversity of age and underlying SMID, I doubt that the data shown in this study could provide adequate evidence to support the clinical usefulness of sway beds for SMID patients.

Author Response

Responses to Reviewer 1

Thank you very much for taking the time to evaluate our manuscript. We have attempted to address all your comments and revise the manuscript accordingly. Changes to the manuscript are indicated in red font. We hope that the revised manuscript meets your approval.

The title was partially changed. “Effect of the sway bed on autonomic response, emotional responses, and muscle hardness in children with severe motor and intellectual disabilities”. This is because this study examined "muscle hardness" rather than "muscle tone.

Comment 1 (In Abstract it is described “by comparing them with an “unrocked”  control group. However, the comparison seemed to have conducted between control and sway stimulus conditions. I also wonder whether there was any difference between “condition” in Table 2 and “group” in Table 3. It is necessary to describe the study design and results to be shown more clearly.)

Response: Thank you for pointing this out. We corrected the inconsistent use of “group” and “condition” in Table 3. Moreover, we revised “unrocked control” to “stimulus(none) control condition” in the Abstract (line 14). The study design was added on line 11.

Comment 2 (The subjects of this study were 10 students attending a special support school who had been diagnosed with SMID, aged between 8 years and 7 months and 18 years and 2 months. I consider that this age range is too wide to examine the effect of the sway bed. The median age is better to be described. )

Response: Thank you for this comment. According to your suggestion, the median age was added because of the wide range of ages (line 72). Further, we added that the target age range of this study is too wide to the limitations section (lines 288-295).

Comment 3 ( The authors showed the characteristics of 10 patients in Table 1. I consider that it would be easier for readers to understand their background characteristics that their summarized data are shown in a table. )

Response: Thank you for this comment. According to your suggestion, we have organized the children’s information into a table (Table 1).

Comment 4 (  It is described in 2.4.2. Emotional responses “After the experiment, the homeroom teacher was given a random time series of the videos of the control and sway stimuli conditions to assess the children’s emotions based on the 8 items of the observation questionnaire and to evaluate them with a 10-point scale”. I wonder whether the assessment using a 10-point scale was reproducible and objective without any clear scale explanation. More detailed descriptions of the assessment of emotional response are necessary. )

Response: Thank you for this valuable comment, We have added a description of the observation questionnaire (lines 105-111). The observational questionnaire used in this study has not been tested for reproducibility. However, such observational questionnaires have been used in previous studies; therefore, we considered that the questionnaire would be suitable for our study. Moreover, the following references were cited as studies in which observational questionnaires have been used:

[20]  Reaves, C.; Angosta, A.D. The relaxation response: influence on psychological and physiological responses in patients with COPD. Appl Nurs Res. 2021, 57, 1–6.

[21]  Gao, Z.; Zhao, L.; Fekete, G.; Katona, G.; Baker, JS.; Gu, Y. Continuous time series analysis on the effects of induced running fatigue on leg symmetry using kinematics and kinetic variables: implications for knee joint injury during a countermovement jump. Front Physiol 2022, 17, 1-12.

Comment 5 (  The core findings of this study were the LF/HF component decreased, and the HF component increased as a result of the sway stimulus, in the frequency analysis of the electrocardiogram, and the subjective emotion rating of Relaxation was significantly larger under sway condition. Due to the wide variety and diversity of age and underlying SMID, I doubt that the data shown in this study could provide adequate evidence to support the clinical usefulness of sway beds for SMID patients. )

Response: Thank you for pointing this out. As mentioned, we were not able to examine the effect of age, underlying disease, and sex on our results. We have highlighted this as a limitation of the study (lines 288-295).

Reviewer 2 Report

First of all, I want to congratulate the authors for the research they have carried out and for submitting the results to peer review.

The article "Effect of the sway bed on autonomic response, emotional responses, and muscle tone in children with severe motor and intellectual disabilities" aims to examine the effects of being rocked in a motorized bed on children with severe motor and intellectual difficulties by examining potential diferences in their autonomic respnse, emotional responses, as well as their muscle tone, comparing with an “unrocked” control group.

It is easy to read and well structured. In the introduction, the concept "severe motor and intellectual disabilities" is defined, evidence on the effect of rocking on several variables relevant to the topic is mentioned, and the importance of the study is described. The methodology is suitable for the type of study, the results answer to the research question and the discussion presents a critical analysis of the results, comparing them with the literature.

However, I will make some comments that should deserve attention from the authors:

1) Only 20% of the bibliography used is less than 5 years old and 48% is more than 10 years old;

2) EKG first appears on line 75, but the acronym is only described on line 97.

3) Figure 1, on line 87, is described as "shows the experimental device", but Figure 1 on page 9 does not match the description. Figure 1 on page 9 corresponds to the description of lines 120–123 which, and here it is correct, ends up saying "(Figure 1)".

4) One of the objectives is to evaluate the effect of the intervention on muscle tone, but to measure this effect, a test that measures muscle hardness is used and, throughout the article, the authors talk about muscle hardness. I ask: are "muscle tone" and "muscle hardness" the same thing?

5) The type of study design is not specified. Is it a randomized control trial?

6) The methodology talks about 10 participants, but the results presentation has 10 participants in the "control conditions" and 10 participants in the "sway stimulus condition". Was it 10 or 20 students? How did you calculate the sample size? How did you do the randomization?

7) How to interpret the values of the questionnaire applied to assess emotional responses? Has the questionnaire been validated?

8) Regarding the methodology for data analysis, why did you use parametric and non-parametric statistics? Given the sample size, don't you think it would have been better to have used nonparametric statistics?

9) Considering the sample size, which is identified as a limitation of the study, it is not ambitious to say that introduction of sway beds in medical institutions, welfare facilities, and schools would provide auxiliary interventions that do not require manpower, which may lead to a reduction of the burden on caregivers looking after children with SMID?

10) Considering the sample size, identified as a limitation of the study, it is not ambitious to say that the introduction of oscillating beds in medical institutions, care institutions and schools would provide ancillary interventions that do not require manpower, which can lead to a reduction in the burden on the  caregivers caring for children with SMID? Another thought that occurs to me: If rocking is proven to relax children and adults, shouldn't the control group have been "non-motorized rocking"?

Author Response

Responses to Reviewer 2

Thank you very much for taking the time to evaluate our manuscript. We have attempted to address all your comments and revise the manuscript accordingly. Changes to the manuscript are indicated in red font. We hope that the revised manuscript meets your approval.

The title was partially changed. “Effect of the sway bed on autonomic response, emotional responses, and muscle hardness in children with severe motor and intellectual disabilities”.This is because this study examined "muscle hardness" rather than "muscle tone.

Comment 1 ( Only 20% of the bibliography used is less than 5 years old and 48% is more than 10 years old)

Response: Thank you for your comment. As mentioned, few of the cited studies have been published within the past 5 years. This is because research on severely handicapped children and sway stimulation is still in its infancy and despite my best efforts, it has been difficult to locate recent references.

Comment 2 (EKG first appears on line 75, but the acronym is only described on line 97.)

Response: Thank you for pointing this out. The text has been corrected (line 76). In addition, the abbreviation was revised from “EKG” to “ECG” throughout the text.

Comment 3 (Figure 1, on line 87, is described as "shows the experimental device", but Figure 1 on page 9 does not match the description. Figure 1 on page 9 corresponds to the description of lines 120–123 which, and here it is correct, ends up saying "(Figure 1)")

Response: As mentioned, the description of Figure 1 on line 87 was an error, and it has been deleted. The text was revised to “The experimental device was an electric sway bed (RHYTHM BED Standard YB, IQuark, Fukuoka, Japan).” (lines 102-103)

Comment 4 ( One of the objectives is to evaluate the effect of the intervention on muscle tone, but to measure this effect, a test that measures muscle hardness is used and, throughout the article, the authors talk about muscle hardness. I ask: are "muscle tone" and "muscle hardness" the same thing?)

Response: Thank you for your comment. In the previous manuscript, there was some confusion about the wording of tone and hardness. Since muscle tone and muscle hardness are different, we did not use the term muscle tone, but unified the term muscle hardness in the revised manuscript. In this study, muscle hardness was defined as “an index that reflects muscle tone at rest.” The following text and reference have been added:

“Muscle hardness testers are easily portable and used to assess muscle relaxation[22].” (lines 136-137)

[22] Sawada, T.; Okawara, H.; Nakashima, D.; Iwabuchi, S.; Matsumoto, M.; Nakamura, M.; Nagura, T. Reliability of Trapezius Muscle Hardness Measurement: A Comparison between Portable Muscle Hardness Meter and Ultrasound Strain Elastography. Sensors (Basel) 2020, 20, 7200. doi: 10.3390/s20247200.

Also, in several sentences, the term muscle tone was changed to muscle hardness.

Comment 5 (The type of study design is not specified. Is it a randomized control trial?)

Response: Thank you for pointing out this omission. We have added the study design to the Abstract (line 11) and section 2.5 of the methods (line 152). The study employed a cross-sectional experimental design.

Comment 6 (The methodology talks about 10 participants, but the results presentation has 10 participants in the "control conditions" and 10 participants in the "sway stimulus condition". Was it 10 or 20 students? How did you calculate the sample size? How did you do the randomization?)

Response: Thank you for pointing this out. We corrected the inconsistent use of “group” and “condition” in Table 3. The study included 10 children. Through the use of a cross-sectional experimental design, each child underwent the control and experimental conditions at random.

Comment 7 (How to interpret the values of the questionnaire applied to assess emotional responses? Has the questionnaire been validated?)

Response: Thank you for this valuable comment. We have added a description of the observation questionnaire (lines 120-126). The observational questionnaire used in this study has not been tested for reproducibility. However, such observational questionnaires have been used in previous studies; therefore, we considered that the questionnaire would be suitable for our study. Moreover, the following references were cited as studies in which observational questionnaires have been used:

[20]  Reaves, C.; Angosta, A.D. The relaxation response: influence on psychological and physiological responses in patients with COPD. Appl Nurs Res 2021, 57, 1–6.

[21]  Gao, Z.; Zhao, L.; Fekete, G.; Katona, G.; Baker, JS.; Gu, Y. Continuous time series analysis on the effects of induced running fatigue on leg symmetry using kinematics and kinetic variables: implications for knee joint injury during a countermovement jump. Front Physiol 2022, 17, 1–12.

Comment 8 (Regarding the methodology for data analysis, why did you use parametric and non-parametric statistics? Given the sample size, don't you think it would have been better to have used nonparametric statistics?)

Response: Thank you for your comment. Given the sample size, nonparametric statistics may be a better choice. However, at the research design stage, we planned to use two-way ANOVA. The use of nonparametric statistics would have precluded the use of two-way ANOVA.

Comment 9 (Considering the sample size, which is identified as a limitation of the study, it is not ambitious to say that introduction of sway beds in medical institutions, welfare facilities, and schools would provide auxiliary interventions that do not require manpower, which may lead to a reduction of the burden on caregivers looking after children with SMID?)

Response: I agree with your comment; thus, the text in question was deleted.

Comment 10 (Considering the sample size, identified as a limitation of the study, it is not ambitious to say that the introduction of oscillating beds in medical institutions, care institutions and schools would provide ancillary interventions that do not require manpower, which can lead to a reduction in the burden on the  caregivers caring for children with SMID? Another thought that occurs to me: If rocking is proven to relax children and adults, shouldn't the control group have been "non-motorized rocking"?)

Response: Thank you for your remarks. Few studies have examined the effects of sway stimulation on children with severe motor and intellectual disabilities. Therefore, in this study, we compared the effects of sway stimulation on children with severe motor and intellectual disabilities by comparing sway stimulation and the control condition of no sway cross-sectionally. I think that “non-motorized rocking” could be a future topic of investigation.

Round 2

Reviewer 1 Report

The authors examined the effects of being rocked in an electric sway bed on 10 students (median age, 15 years, 8-18 years) with severe motor and intellectual difficulties by examining potential differences in their autonomic and emotional responses and their muscle hardness. There findings were significant intervention-related effects of the sway bed were observed on heart rate variability but not on muscle hardness, hip ROM. Significant change  was observed only in Relaxation in the subjective emotion ratings.

First of all, I wonder how the authors determined the sample numbers. If the authors had determined the adequate sample size calculated based on data from a pilot study, more novel findings could have found. I would recommend the authors to consult statisticians to how to determine  the adequate sample size and increase the sample numbers.

I wonder why the data that a significant difference was noted in the “Relax” item" in the subjective emotion ratings were not included in Abstract.

I wonder what is the original purpose of the the electric sway bed. The photo of the instrument, the movie of students' swaying if possible, would be appreciated with more details of this instrument.

Author Response

Responses to Reviewer 1

Thank you very much for taking the time to evaluate our manuscript. We have attempted to address all your comments and revise the manuscript accordingly. Changes to the manuscript are indicated in red font. We hope that the revised manuscript meets your approval.

Comment 1 (First of all, I wonder how the authors determined the sample numbers. If the authors had determined the adequate sample size calculated based on data from a pilot study, more novel findings could have found. I would recommend the authors to consult statisticians to how to determine the adequate sample size and increase the sample numbers.)

Response: Thank you for this comment. Sample size calculations were performed using the GPower software version 3.1 (G*Power: Universität Düsseldorf, Düsseldorf, Germany; 2010e2016) with “a priori calculation”. This is an analysis method to calculate sufficient sample sizes to achieve adequate power prior to the research study. For this calculation, we used an alpha value of 0.05, an effect size of 0.8, and a power of 80% and identified that a sample size of 30 participants was sufficient. While we aimed to recruit 30 participants, due in part to COVID-19, it was difficult to reach this target. Our study sample size was further limited due to the exclusion of children with a GMFCS level of I-IV. The need for a larger sample size will be considered in future studies. Thus, the small sample size is emphasized and mentioned as a limitation in this study. Also, although the sample size was small this time, the Shapiro-Wilk test confirmed that the distribution was normal, so we have added this information to the manuscript as well.

Our revisions to the manuscript in response to this comment are as follows:

“Children with the following conditions were excluded from the study: Gross Motor Function Classification System (GMFCS) level I-IV, aspiration pneumonia, respiratory disease, heart disease, susceptibility to motion sickness, and sensitive skin (to avoid irritation from electrocardiogram [ECG] electrodes).” (Line 75-79)

“Initially, 13 were included in the study; however, three children with GMFCS level IV were excluded, resulting in a final total of 10 children participants.” (Line 79-80)

“Normality of the data distributions for HRV (LF/HF , HF), muscle hardness (N), and hip ROM (cm) was confirmed by the Shapiro-Wilk test.” (Line 226-227)

“Data of HRV (LF/HF , HF), muscle hardness (N), and hip ROM (cm) were confirmed to be normally distributed by the Shapiro-Wilk test.” (Line 242-243)

“This study has limitations regarding the sample size and age range. Owing to the small sample size of 10 children, we were unable to examine relationships between sex, age, and the degree of change in the data before and after the sway stimulation. Since SMID has many underlying causes, these relationships should be investigated using large samples. Moreover, a wide age range of children were included in this study, which may have affected the results. Future studies should include subjects with a narrower age range.” (Line 308-314)

Comment 2 (I wonder why the data that a significant difference was noted in the “Relax” item" in the subjective emotion ratings were not included in Abstract. )

Response: This was an oversight on our part so we thank you for bringing this to our attention. The manuscript has been revised accordingly as follows:

“Significant intervention-related effects were observed on heart rate variability and the “Relax” item of the emotional response indicators but not on muscle hardness or hip ROM.” (Line 18-20)

Comment 3 ( I wonder what is the original purpose of the the electric sway bed. The photo of the instrument, the movie of students' swaying if possible, would be appreciated with more details of this instrument.)

Response: Thank you for this comment. We have added infomraton regarding the original purpose of the sway bed in the manuscript as follows:

“The sway bed is a welfare device that was created at the request of parents to provide muscle tension relief and psychological relaxation to children with SMID. Although it was created for this purpose, its effectiveness has not yet been fully verified.” (Line 53-55)

We have also added a photo and description of the sway bed as Figure 1 in the manuscript:

“Figure 1 shows the electric sway bed (RHYTHM BED Standard YB, IQuark, Fukuoka, Japan).”(Line 105-106)

Reviewer 2 Report

Thank you for responding to the suggestions and questions raised in the 1st round of review. The changes you made have improved the quality of the manuscript. I have no further comments to make. I hope that a study with more participants will soon be published.

Author Response

Thank you for your review.